TECHNICAL RELEASE

# Chevreul: an R bioconductor package for exploratory analysis of full-length single cell sequencing

Kevin Stachelek[1,2,*], Bhavana Bhat[1] and David Cobrinik[1,3,4,5,*]

1  The Vision Center, Department of Surgery, and Saban Research Institute, Children's Hospital Los Angeles, Los Angeles, CA 90027, USA
2  Cancer Biology and Genomics Program, Keck School of Medicine, University of Southern California, Los Angeles, CA 90033, USA
3  Department of Cancer Biology, Keck School of Medicine, University of Southern California, Los Angeles, CA 90033, USA
4  Norris Comprehensive Cancer Center, Keck School of Medicine, University of Southern California, Los Angeles, CA 90033, USA
5  USC Roski Eye Institute, Department of Ophthalmology, Keck School of Medicine, University of Southern California, Los Angeles, CA 90033, USA

## ABSTRACT

Chevreul is an open-source R Bioconductor package and interactive R Shiny app for processing and visualising single-cell RNA sequencing (scRNA-seq) data. Chevreul differs from other scRNA-seq analysis packages in its ease of use, capacity to analyze full-length RNA sequencing data for exon coverage and transcript isoform inference, and support for batch correction. Chevreul enables exploratory analyses of scRNA-seq data using Bioconductor SingleCellExperiment objects (or converted Seurat objects), including batch integration, quality control filtering, read count normalization and transformation, dimensionality reduction, clustering at a range of resolutions, and cluster marker gene identification. Processed data can be visualized in the R Shiny app. Gene or transcript expression can be visualized using PCA, tSNE, UMAP, heatmaps, or violin plots; differential expression can be evaluated with several statistical tests. Chevreul also provides accessible tools for isoform-level analyses and alternative splicing detection. Chevreul empowers researchers without programming experience to analyze full-length scRNA-seq data.

**Availability & implementation:** Chevreul is implemented in R, and the R package and integrated Shiny application are freely available at https://github.com/cobriniklab/chevreul with constituent packages hosted on Bioconductor at https://bioconductor.org/packages/chevreulProcess, https://bioconductor.org/packages/chevreulPlot, and https://bioconductor.org/packages/chevreulShiny.

**Submitted:**  20 December 2024

* Corresponding authors. E-mail: stachele@usc.edu; dcobrinik@chla.usc.edu

Preprint submitted at https://doi.org/10.1101/2025.05.27.656486

**Subjects**  Software and Workflows, Bioinformatics, Biomedical Science

## STATEMENT OF NEED

Exploratory data analysis is an important step in single-cell RNA sequencing (scRNA-seq) studies, which is made more challenging by the complexity of sequencing outputs. Widely used scRNA-seq analysis toolkits, including Seurat (RRID:SCR_016341) [1] for R and Scanpy (RRID:SCR_018139) [2] for Python, enable flexible, reproducible analysis and codify best-practices [2, 3]. However, these command-line toolkits can be challenging for biologists without experience with relevant programming or statistics concepts. Several interactive tools for single-cell data analysis do not require such experience, including CELLxGENE (RRID:SCR_021059), scClustViz, and Cerebro [4–6]. However, these tools are primarily

designed to analyze droplet-based sequencing methods that yield 3′- or 5′-end short-read sequences; also, they do not take advantage of Smart-seq-based or other deep, full-length RNA sequencing datasets, which could provide greater insights into transcript isoforms and cell states. While the BioConductor iSEE ecosystem [7] enables first-class analyses of preprocessed full-length RNA sequencing datasets and associated visualizations via extension packages, it does not prioritize functions for processing such data within an interactive Shiny app.

Chevreul is an open-source R Bioconductor package for exploratory analyses of scRNA-seq data, including deep, full-length scRNA-seq data, processed in the SingleCellExperiment Bioconductor or Seurat formats. The package simplifies scRNA-seq data analysis using a set of R Shiny apps to visualize scRNA-seq datasets with interactive plots. Chevreul is distinct from other interactive tools for single-cell data analysis in its orientation toward full-length sequencing data captured by Smart-seq-based cDNA synthesis, followed by short-read sequencing [8], which necessitates analyses such as measuring isoform-specific expression and transcript exon coverage. Chevreul is also notable for its support for batch integration without requiring a coding environment. Batch integration is a primary concern for laboratories working with rare cell types and clinical samples with limited cell numbers, which necessitate multiple rounds of just-in-time processing and introduce technical batch effects. Furthermore, Chevreul includes easy-to-execute batch integration and data preprocessing pipelines with sensible default settings. Using the Chevreul R Shiny app, researchers can analyze data from multiple experiments within a common framework, greatly reducing the time needed to generate publication-ready plots, novel hypotheses, and insights from scRNA-seq experiments. Chevreul incorporates automated testing using the *testthat* package [9]. It can be easily installed on a Windows, Mac, or UNIX-like operating system from the current Bioconductor release and will be maintained and updated according to biannual Bioconductor releases. The package was inspired by the color theorist Michel-Eugène Chevreul and the optical illusion of the same name [10].

## IMPLEMENTATION

### Processing workflow

Chevreul is an R package (actually, a meta-package) for processing and integration of scRNA-seq data from cDNA end-counting, full-length short-read or long-read protocols, and an R Shiny app for easy visualization, formatting, and analysis. Chevreul installs and loads three Bioconductor packages: *chevreulProcess* for processing, *chevreulPlot* for plotting, and *chevreulShiny* for loading and interactive analysis of processed scRNA-seq data (including full-length scRNA-seq) as SingleCellExperiment objects. All functionality contained in the three constituent packages is accessible directly from Chevreul.

Chevreul enables the processing of an scRNA-seq dataset starting with a SingleCellExperiment object [11] constructed from a table of cell metadata with experimental variables, such as treatment status, and a raw gene-by-cell or transcript-by-cell count matrix output from common tools, such as *tximport* (RRID:SCR_016752) [12], *CellRanger* [13], or *STAR* (RRID:SCR_004463) [14]. If data are derived from full-length scRNA-seq protocols, transcripts may be quantified using alignment-free methods best used with well-annotated transcriptomes (Salmon (RRID:SCR_017036), Kallisto (RRID:SCR_016582)), alignment-based methods best used to



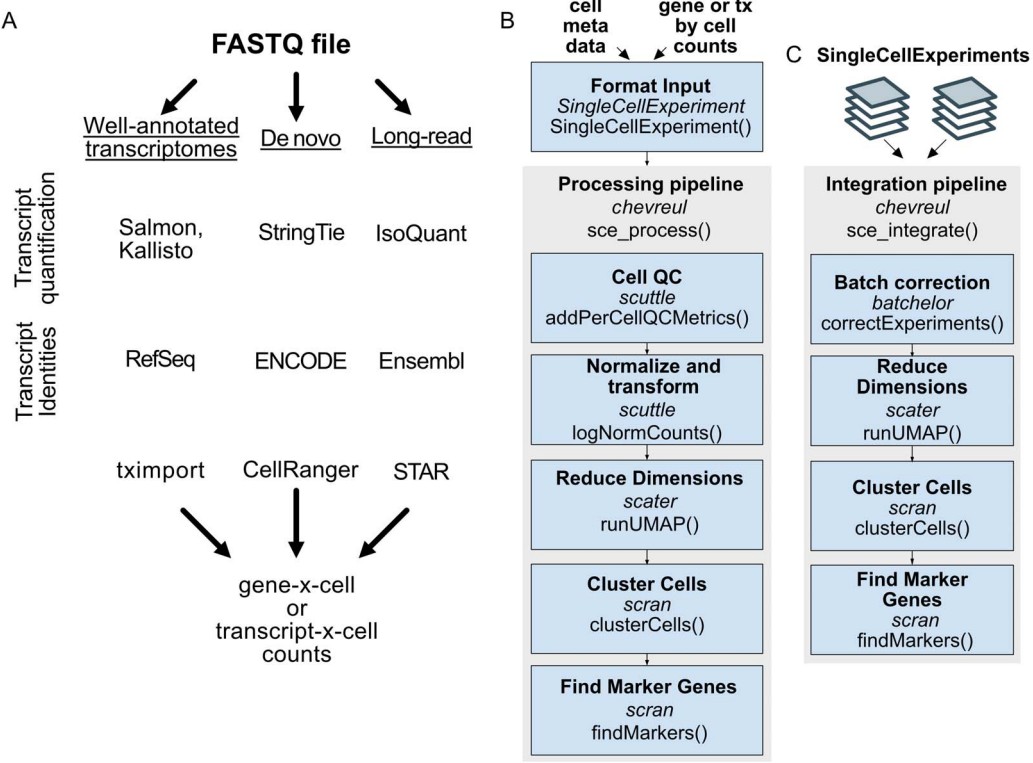

**Figure 1.** **Full-length scRNAseq processing workflows and subsequent analyses with Chevreul.**
(A) Example of workflows preceding processing and quantitation of full-length scRNA-seq datasets. (B) Processing pipeline for quality control filtering, read count normalization and transformation, dimensionality reduction, clustering, and marker gene identification. (C) Pipeline for batch integration and processing. Pipeline steps indicate *R package dependency* and their specific function. See Methods for open-source software sources and the Chevreul package for default settings and code.

detect novel isoforms (StringTie2), or long-read methods for use with long-read sequencing data such as Oxford Nanopore or PacBio technologies (IsoQuant) [15–18].

Isoform identities can be specified according to any version of the three major gene annotation databases [19–21] used in upstream transcript mapping and quantitation (Figure 1A). Reads are then imported as separate transcript and gene assays, where gene read counts are the sum of the constituent transcript counts.

In chevreulProcess, standardized functions prefixed with 'sce_' enable the processing and integration of datasets. Initially, single datasets can be processed with a simple pipeline function, *sce_process()*, which allows (i) quality control filtering by minimum expression and ubiquity to exclude empty droplets or degraded cells, (ii) normalization by library-size scaling and log transformation, (iii) dimensionality reduction by principal component analysis (PCA), t-Distributed Stochastic Neighbor Embedding (tSNE), and Uniform Manifold Approximation and Projection (UMAP), (iv) Louvain clustering at a range of resolutions, and (v) cluster marker gene or marker transcript identification (Figure 1B). Sensible default values are provided for all processing steps, although individual steps can be adjusted at any stage. A second pipeline function, *sce_integrate()*, enables the integration of a list of SingleCellExperiment objects and is followed by batch-corrected processing, including dimensional reduction and clustering (Figure 1C). *sce_integrate()* also implements batch

correction of scRNA-seq datasets using the *batchelor* package [22]. Default settings rely on the *batchelor correctExperiments()* function to preserve the pre-existing data and metadata from input objects in the corrected output.

Chevreul enables analyses of Seurat-derived scRNA-seq objects only after their conversion to the SingleCellExperiment format. This may be preferred when a Seurat object was previously generated or when SingleCellExperiment datasets are prepared using different bioinformatic approaches or dimensional reduction parameter settings. For example, initially preparing a Seurat object may be desired to take advantage of alternative batch integration methods obtained using the *Seurat* package [1], which yield different normalized data input for dimension reduction and clustering compared to objects generated with the Chevreul function *sce_integrate()*. Previously generated Seurat V5 objects can be converted to SingleCellExperiment objects using the package *seuFLViz* [23] and further examined using Chevreul.

The recommended minimum hardware requirements for running Chevreul include 16 GB of RAM. However, for larger datasets or more complex analyses, 64 GB or more is advisable. Additionally, having multiple cores can be beneficial for parallel processing. As the number of cells increases, so do the hardware requirements. For instance, a full-length scRNA-seq dataset with ~800 cells and an average of $3.75 \times 10^6$ reads per cell can be analyzed with 8 GB of RAM. For larger datasets or highly complex analyses, 64–128 GB of RAM can be beneficial. Runtime and memory scaling, for an example dataset, are reported for an Ubuntu 20.04 system with an 8-core Intel i7 CPU and 69 GB of RAM (Figure 2).

## Interactive analysis

To demonstrate the use of Chevreul, we provide a sample analysis of a publicly available Smart-seq-based scRNA-seq dataset of the developing human retina (GSE207802), which was previously evaluated with a Shiny app based on the *Seurat* R package [24] and is available in a preprocessed SingleCellExperiment object in the *chevreuldata* package. Chevreul documentation includes a full workflow with example vignettes, including an annotated guide for the Shiny app, a walk-through of all plotting functions, a getting-started guide for troubleshooting installation, links to relevant educational resources, background information on isoform-level analyses, and basic execution instructions.

Starting with a processed SingleCellExperiment, the Chevreul Shiny app enables the plotting of feature expression and cell metadata variables for visual analysis and data exploration. Dynamic linking between plots ensures a consistent view of a loaded dataset, allowing for the discovery of unrecognized relationships between experimental variables. Interactive plots are accessed on sidebar tabs on the dashboard (Figure 3A). The Overview Plots tab allows the visualization of embeddings in PCA, tSNE, or UMAP overlaid with cell metadata or feature (i.e., gene or transcript) expression (Figure 3B). Dimensional reduction parameters, such as the UMAP input dimensions and the UMAP minimum distance, can be tuned to adjust the embedding appearance. Alternatively, UMAP embeddings generated using Seurat can be imported into Chevreul and analyzed with the Chevreul Shiny app (Figure 4A). This can yield different, albeit similar, clusters and dimensional reduction plots due to the different batch integration methods used by Seurat integration [1] and Bioconductor mnnCorrect [22]. Additionally, UMAPs are also affected by the selected random seed [25]. An additional section of the Overview Plots window displays read or

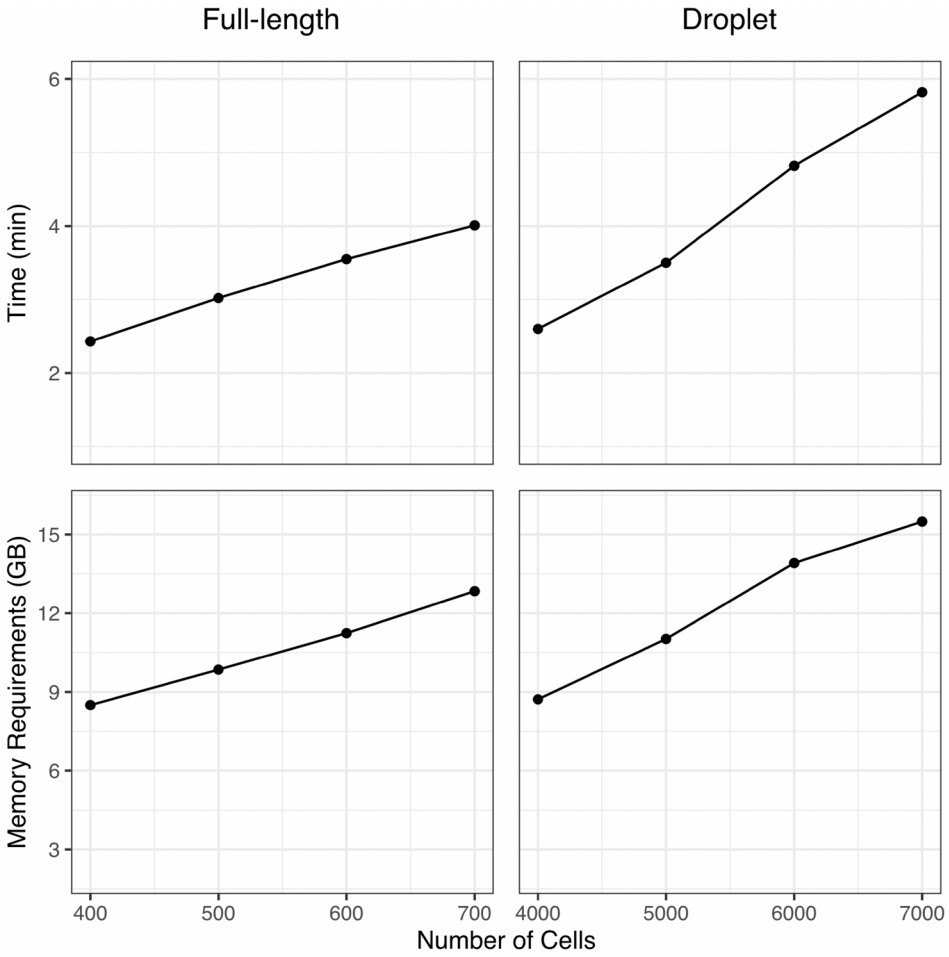

**Figure 2.** **System time and memory used for initial processing of a full-length and droplet-based SingleCellExperiment.**
The example compares full-length scRNA-seq (794 cells and 8,277 mean genes detected) with simulated droplet scRNA-seq (8,000 cells and 8,994 mean genes detected) datasets processed using *sce_process()* with default Chevreul settings. Genes defined using ensemble build 87. Benchmarking was carried out on an Ubuntu 20.04 system with an 8-core Intel i7 CPU and 69 GB of RAM.

unique-molecular-identifier count-histograms, which can be colored based on metadata variables of interest, such as the Louvain cluster identity (Figure 3C). Finally, a Clustering Tree plot is provided to visualize the relationship between Louvain clusters at a range of resolutions using the *clustree* package [26] (Figure 3D).

The Violin tab displays customizable violin plots of gene expression organized by cell metadata, such as cell cluster, sample age, sample genotype, or multiple variables in combination (Figure 5A). The Differential Expression tab allows differential expression analyses between groups specified by cell metadata or graphical selection from a dimensionally reduced plot. Resulting gene lists can be exported, and volcano plots colored and labeled for genes surpassing desired *p*-values and fold-change thresholds (Figure 5B). The Find Markers tab allows the plotting of marker features, such as genes or transcripts that are specific to cluster identities or other cell metadata, and are defined based on a Wilcoxon rank-sum test. A user-assigned number of marker genes per cell group can be

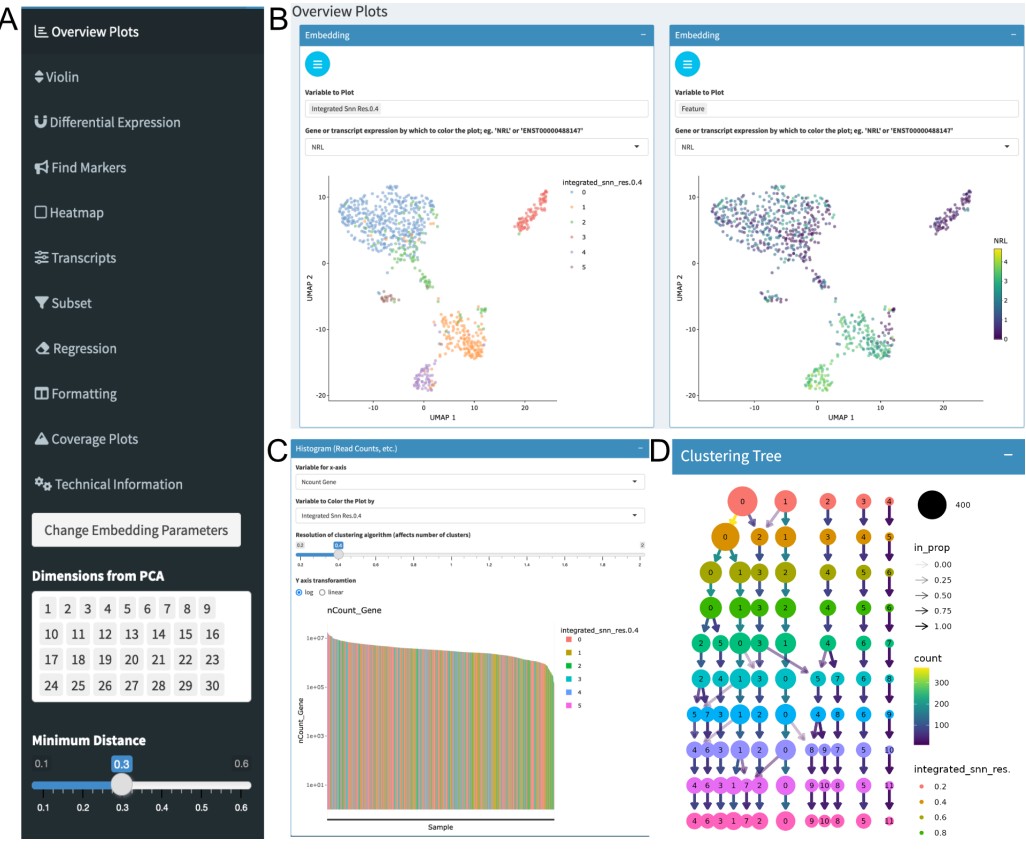

**Figure 3.** **Analysis overview plots.**
(A) Sidebar showing Shiny app tabs. (B) UMAP embedding overlaid with metadata for cluster cell type or expression of *NRL*. (C) Read count histogram colored by cluster identity. (D) Clustering tree at multiple resolutions.

specified and those meeting adjusted *p*-value and log fold-change thresholds displayed (Figure 5C). The Heatmaps tab allows plotting via complexHeatmap (RRID:SCR_017270) [27] to display selected lists of genes or transcripts (Figure 5D).

If the input SingleCellExperiment contains both gene and transcript counts, the Transcripts tab allows the display of a gene's constituent inferred transcripts as stacked bar plots for different groups (e.g., cluster or sample age) (Figure 6A) or overlaid on PCA, tSNE, and UMAP embeddings (Figures 6B and 4B). The Coverage tab allows plotting of exon read coverage with log or linear *y*-axis scales using the *wiggleplotr* R package [28] (Figure 6C). All plots shown are interactive using Plotly (RRID:SCR_013991) [29], which allows plots to be viewed at various zoom levels and allows cells to be identified by placing the cursor over dots (in embeddings) or over lines (in histograms). User-customized plots can be downloaded in raster (.png) or vector (.svg, .pdf) image formats for further analysis and presentation.

The Subset tab enables iterative analyses of data subsets defined by an uploaded delimited file or by using the cursor to select groups of cells from a dimensionally reduced plot. Subsetting of single or batch-integrated data triggers the renewal of all relevant preprocessing steps including dimensional reduction, clustering, and marker genes, as well as integration based on a 'batch' variable. After subsetting, a reset option can be triggered



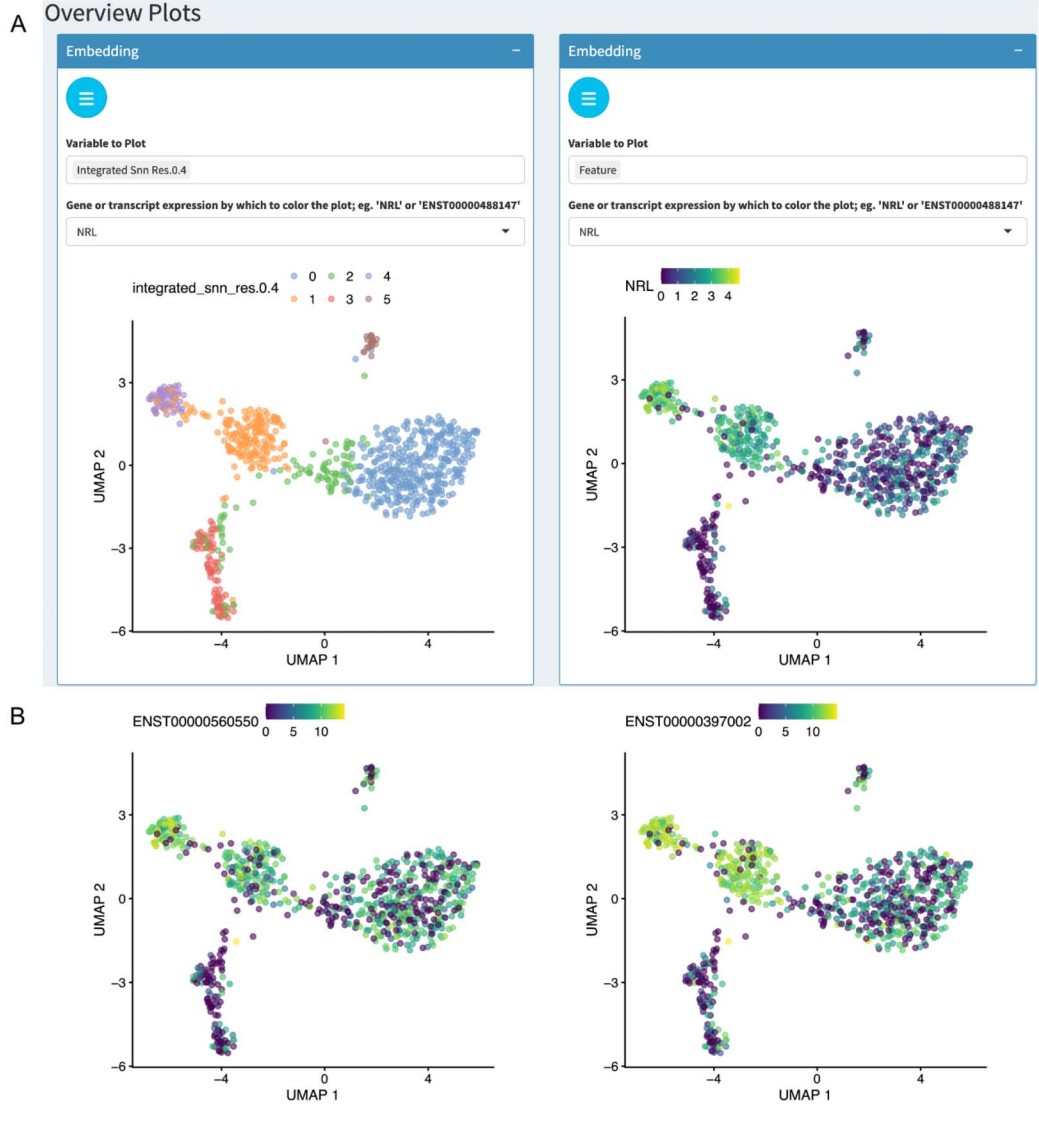

**Figure 4.  Analysis overview plots with dataset processed with Seurat.**
(A) UMAP overlaid with metadata for cluster cell type (left) or expression of *NRL* (right). (B) Transcript-specific plots processed with Seurat. Expression of *NRL* transcripts ENST00000397002 and ENST00000560550 aggregated gene expression.

within the subset tab to return data to an initial state. In the Regression tab, confounding cell-cycle effects can be regressed using cyclone [30]. The Formatting tab displays the original cell metadata and allows reformatting by in-app editing or by uploading a minimal file with cell identifiers and any new cell metadata to be added. The resulting reformatted tables can be exported as .csv files. All analysis steps, including software versions, are specified in a convenient Technical Information tab.

## DISCUSSION

Chevreul is an open-source R package and R Shiny app that empowers researchers without programming experience to analyze scRNA-seq data. Chevreul allows processing,



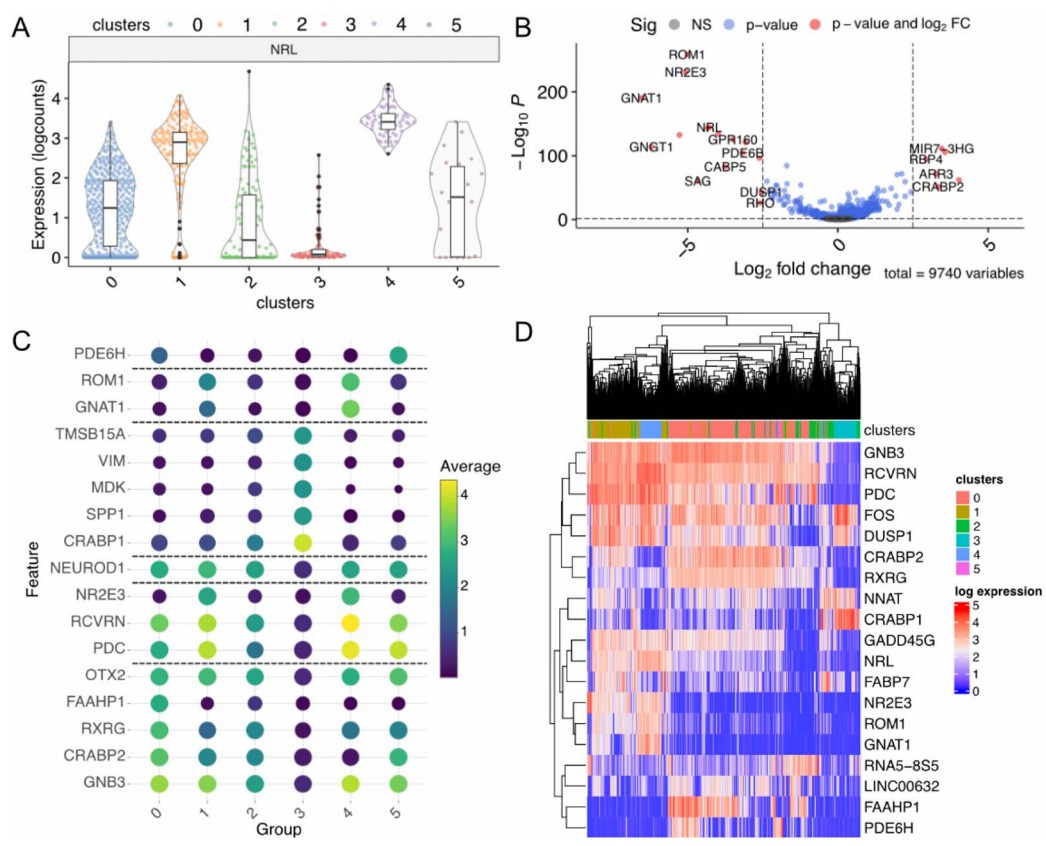

**Figure 5.  Gene expression plots.**
(A) Violin plot of expression colored by cell metadata. (B) Volcano plot after differential expression. (C) Cluster marker genes. (D) Heatmap of highly variable genes.

visualization, and interactive analysis in a standardized framework. The built-in Shiny app includes plotting functions specifically designed for full-length scRNA-seq data, including exon coverage and inferred isoform usage summarized according to the sample characteristics represented in the cell metadata. Inferring isoform usage, in turn, may reveal cell-type-specific transcript isoform use and enable distinctions between closely related states that are not visible with gene expression alone [24]. The Chevreul application includes extensive guidance materials and allows users to visualize a wide range of parameters, enabling transparent and reproducible scRNA-seq analyses.

## METHODS

The open-source software used in the Chevreul package includes:

| Software | Version |
| --- | --- |
| R | 4.4.0 |
| alabaster.base | 1.4.2 |
| batchelor | 1.19.1 |
| BiocManager | 1.30.22 |
| BiocStyle | 2.31.0 |

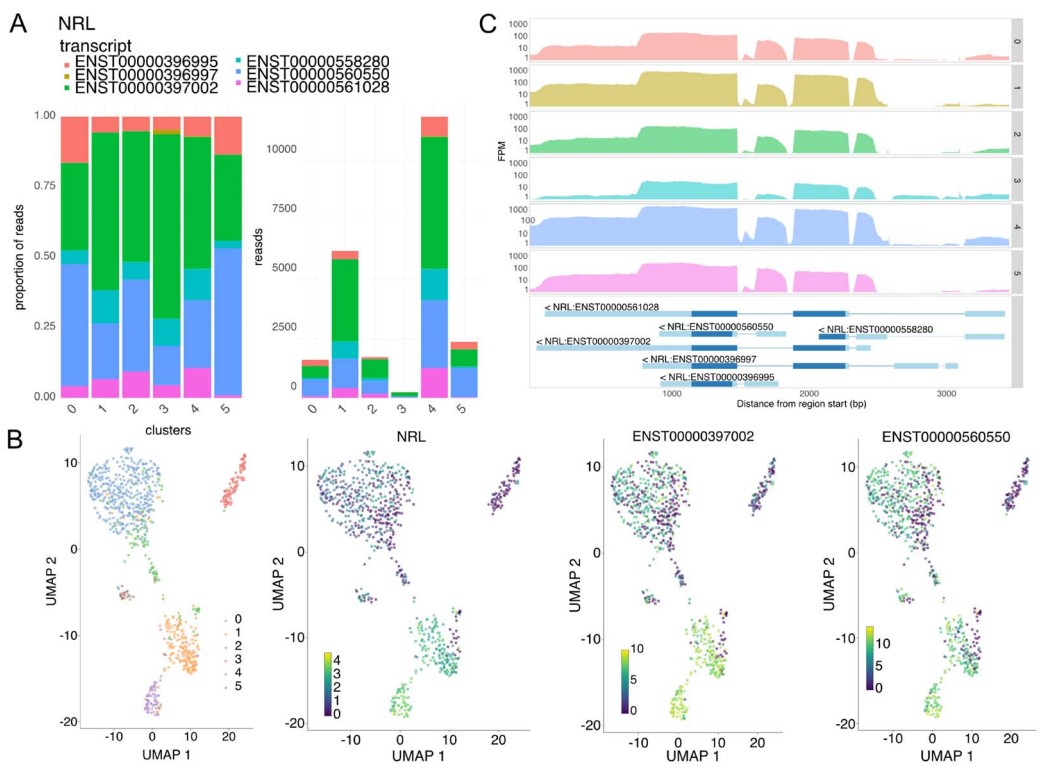

**Figure 6. Transcript-specific plots.**
(A) Transcript composition by cell group as stacked bar plots. (B) *NRL* expression of constituent transcripts and aggregated gene expression. (C) Coverage of gene regions colored by cell metadata using wiggleplotr [28].

| | |
|---|---|
| BiocVersion | 3.19.1 |
| biomaRt | 2.59.1 |
| bluster | 1.13.0 |
| circlize | 0.4.16 |
| cluster | 2.1.6 |
| clustree | 0.5.1 |
| ComplexHeatmap | 2.15.4 |
| DataEditR | 0.1.6 |
| DBI | 1.2.2 |
| DT | 0.32 |
| EnhancedVolcano | 1.21.0 |
| EnrichmentBrowser | 2.33.1 |
| EnsDb.Hsapiens.v86 | 2.99.0 |
| EnsDb.Mmusculus.v79 | 2.99.0 |
| ensembldb | 2.27.1 |
| ExperimentHub | 2.11.1 |
| GenomicFeatures | 1.55.3 |
| ggplotify | 0.1.2 |
| ggpubr | 0.6.0 |
| ggraph | 2.1.0 |

| | |
|---|---|
| glue | 1.7.0 |
| harmony | 1.2.0 |
| megadepth | 1.13.0 |
| patchwork | 1.2.0.9000 |
| plotly | 4.10.4 |
| rappdirs | 0.3.3 |
| rmarkdown | 2.25 |
| RSQLite | 2.3.5 |
| S4Vectors | 0.41.3 |
| scales | 1.3.0 |
| scater | 1.31.2 |
| scran | 1.31.0 |
| scuttle | 1.13.0 |
| shiny | 1.8.0 |
| shinydashboard | 0.7.2 |
| shinyFiles | 0.9.3 |
| shinyhelper | 0.3.2 |
| shinyjs | 2.1.0 |
| shinyWidgets | 0.8.1 |
| SingleCellExperiment | 1.25.0 |
| SummarizedExperiment | 1.33.3 |
| waiter | 0.2.5 |
| whisker | 0.4.1 |
| wiggleplotr | 1.27.0 |

## AVAILABILITY OF SOURCE CODE AND REQUIREMENTS

The Chevreul package is freely available on GitHub https://github.com/cobriniklab/chevreul with constituent packages hosted on Bioconductor [31–33]. A companion package for processing and displaying Seurat objects is available at https://github.com/cobriniklab/seuFLViz. The software is registered in the bio.tools database at biotoolsID (biotools:chevreul) and the Scicrunch.org database at Research Resource Identification Initiative ID (RRID:SCR_026966).

- Project name: Chevreul
- Project home page: https://cobriniklab.github.io/chevreul/
- Operating system(s): Platform independent
- Programming language: R 4.4.0
- License: MIT
- RRID:SCR_026966
- biotoolsID: biotools:chevreul.

## DATA AVAILABILITY

A test dataset formatted as a SingleCellExperiment object can be found at https://github.com/cobriniklab/chevreuldata. Test data is also available in the GigaDB repository [34].

## ABBREVIATIONS

PCA: principal component analysis; scRNA-seq: single-cell RNA sequencing; tSNE: t-Distributed Stochastic Neighbor Embedding; UMAP: Uniform Manifold Approximation and Projection.

## DECLARATIONS

### Ethical approval

Not applicable.

### Competing interests

The authors declare that they have no competing interests.

### Authors' contributions

KS: conceptualization, investigation, formal analysis, software, methodology, validation, data curation, writing – original draft preparation, writing – review and editing; visualization, project administration; BB: investigation, software, data curation, visualization; DC: supervision, project administration, writing – review and editing, funding acquisition.

### Funding

National Institutes of Health grant R01EY026661 (DC), National Institutes of Health grant R01CA137124 (DC), Research to Prevent Blindness (unrestricted grant to USC Dept. of Ophthalmology), Larry and Celia Moh Foundation (DC), Neonatal Blindness Research Fund (DC), A.B. Reins Foundation (DC), Knights Templar Eye Foundation (DC).

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
