## [Editor Report]

Editor’s AssessmentThis paper presents Chevreul, a new open-source R Bioconductor (meta-)package for processing and integration of scRNA-seq data from cDNA end-counting, full-length short-read or long-read protocols. Alongside a R Shiny app for easy visualization, formatting, and analysis for exploratory analyses of scRNA-seq data processed in the SingleCellExperiment Bioconductor or Seurat formats. The name of the tool is inspired by the colour theorist Michel-Eugène Chevreul and the optical illusion of the same name. To demonstrate the use of Chevreul, the authors provide a sample analysis, which helps to demonstrate how users can visualize a wide range of parameters, enabling transparent and reproducible scRNA-seq analyses. Peer review also pushing the author to provide extensive guidance materials to assist with use. Being implemented in R, the R package and integrated Shiny application are freely available under an open-source MIT license in Bioconductor and their GitHub page here: https://github.com/cobriniklab/chevreulEditor’s AssessmentThis paper presents Chevreul, a new open-source R Bioconductor (meta-)package for processing and integration of scRNA-seq data from cDNA end-counting, full-length short-read or long-read protocols. Alongside a R Shiny app for easy visualization, formatting, and analysis for exploratory analyses of scRNA-seq data processed in the SingleCellExperiment Bioconductor or Seurat formats. The name of the tool is inspired by the colour theorist Michel-Eugène Chevreul and the optical illusion of the same name. To demonstrate the use of Chevreul, the authors provide a sample analysis, which helps to demonstrate how users can visualize a wide range of parameters, enabling transparent and reproducible scRNA-seq analyses. Peer review also pushing the author to provide extensive guidance materials to assist with use. Being implemented in R, the R package and integrated Shiny application are freely available under an open-source MIT license in Bioconductor and their GitHub page here: https://github.com/cobriniklab/chevreul

---

## [Reviewer Report]

Indicate in the comments box below whether you are happy with the changes made or if the manuscript is unacceptable.Comments on revised manuscriptI am happy with the revision and author have fully addressed my concerns.

---

## [Reviewer Report]

Indicate in the comments box below whether you are happy with the changes made or if the manuscript is unacceptable.Comments on revised manuscriptI have carefully reviewed the revised manuscript and am satisfied that all my comments have been adequately addressed. The authors have resolved the software errors reported in the original submission by updating the relevant shiny app modules. They have also enhanced the package documentation to assist users without programming experience in installing and using Chevreul. In the manuscript itself, the authors have provided detailed responses and explanations to each of my points. Overall, they have addressed all of my comments thoroughly. That said, a few minor issues remain in the manuscript (revised version with tracked changes) that should be corrected to ensure consistency with academic publishing standards and to help readers better learn how to use Chevreul: 1. On line 52, the placeholder “(doi reference for Shayler et al. data to be provided)” appears—did the authors forget to insert the citation or data link? 2. On line 96, would it be more appropriate to replace “SingleCellExperiments” with “SingleCellExperiment objects”? 3. On line 119, please add a space so that “databases[19–21]used” reads “databases [19–21] used.” 4. For consistency, should the second occurrence of “batchelor” on line 132 be italicized? 5. The Chevreul link is already cited in the “Availability & Implementation” section and need not be repeated in the Figure 1 legend. 6. On line 184, the gene symbol “NRL” should be set in italic Latin script. 7. On the GitHub page (https://github.com/cobriniklab/chevreul), the phrase “A demo with a developing human retina scRNA-seq dataset from Shayler et al. is available here” points to an inaccessible web demo. Restoring this demo in a future update would greatly facilitate experimental biologists in learning and using Chevreul.

---

## [Reviewer Report]

Reviewer name and names of any other individual's who aided in reviewerDr. Luyi Tian and Dr. Hongke PengDo you understand and agree to our policy of having open and named reviews, and having your review included with the published manuscript. (If no, please inform the editor that you cannot review this manuscript.)YesIs the language of sufficient quality?YesPlease add additional comments on language quality to clarify if neededIs there a clear statement of need explaining what problems the software is designed to solve and who the target audience is? YesAdditional CommentsThus, the statement of need is well-defined, addressing both the problem (complexity of scRNA-seq data analysis without programming skills) and the intended audience (non-programming researchers in the field).Is the source code available, and has an appropriate Open Source Initiative license <a href="https://opensource.org/licenses" target="_blank">(https://opensource.org/licenses)</a> been assigned to the code?YesAdditional CommentsAs Open Source Software are there guidelines on how to contribute, report issues or seek support on the code?YesAdditional CommentsIs the code executable?Unable to testAdditional CommentsIs installation/deployment sufficiently outlined in the paper and documentation, and does it proceed as outlined?YesAdditional CommentsIs the documentation provided clear and user friendly?YesAdditional CommentsIs there enough clear information in the documentation to install, run and test this tool, including information on where to seek help if required?Additional CommentsIs there a clearly-stated list of dependencies, and is the core functionality of the software documented to a satisfactory level?YesAdditional CommentsHave any claims of performance been sufficiently tested and compared to other commonly-used packages? NoAdditional CommentsIs test data available, either included with the submission or openly available via cited third party sources (e.g. accession numbers, data DOIs)?Additional CommentsAre there (ideally real world) examples demonstrating use of the software? YesAdditional CommentsIs automated testing used or are there manual steps described so that the functionality of the software can be verified?Additional CommentsAny Additional Overall Comments to the AuthorThis study provides Chevreul, a Bioconductor package, for analysis and visualization of single-cell sequencing data. This package contains a shinny app. It also provide the functions which implemented by a set of bioconductor packages for standard scRNA-seq analysis to generate the necessary input of the shinny app. I believe that this app can provide an additional option for researchers who work with single-cell data. However, there might be a few comments need addressing. While the title emphasizes "exploratory analysis of full-length single-cell sequencing," the authors do not explicitly mention the analysis full-length data (e.g., isoform detection or quantification). For instance, the “sce_process(...)” pipeline figure lacks specific steps addressing full-length sequencing workflows. To strengthen this claim, the authors might need to mention/summarize the methods for isoform detection and quantification, for both annotated and novel ones. It would be better to specify recommended tools for transcript-level analysis (e.g., transcript assembly or differential isoform usage) that integrate with Chevreul's visualization features. Meanwhile, The manuscript focuses on Smart-seq as the representative full-length method. It might also be helpful to discuss other full-length methods such as ONT nanopore sequencing or PacBio, in aspect of data processing, transcript assembly, de novel usage or potential challenges in adapting Chevreul to these platforms, etc. There is another minor suggestion. Functions mentioned in the text and Figure 1 (e.g., “sce_process”, “sce_integrate”) should include parentheses (e.g., “sce_process()”) to align with R syntax conventions and clarify their roles as package functions.RecommendationMajor Revisions

---

## [Reviewer Report]

Reviewer name and names of any other individual's who aided in reviewerTianhang LvDo you understand and agree to our policy of having open and named reviews, and having your review included with the published manuscript. (If no, please inform the editor that you cannot review this manuscript.)YesIs the language of sufficient quality?YesPlease add additional comments on language quality to clarify if neededIs there a clear statement of need explaining what problems the software is designed to solve and who the target audience is? YesAdditional CommentsChevreul provides tools for exploratory analysis of single-cell data and offers essential tools for the analysis and visualization of single-cell full-length transcriptomes. In several sections of the article, the authors discuss the key computational challenges addressed by this software. However, in the abstract, they need to emphasize the advantages of Chevreul in single-cell full-length transcript analysis (the current version lacks sufficient description). In the "Statement of Need" section, the authors could also highlight the limitations of existing single-cell full-length transcript analysis tools and introduce the advantages of Chevreul in this regard.Is the source code available, and has an appropriate Open Source Initiative license <a href="https://opensource.org/licenses" target="_blank">(https://opensource.org/licenses)</a> been assigned to the code?YesAdditional CommentsAs Open Source Software are there guidelines on how to contribute, report issues or seek support on the code?YesAdditional CommentsIs the code executable?YesAdditional CommentsAlthough some functionalities can be implemented in Shiny, there are still noticeable bugs. For example, the current version encounters an error when plotting Coverage graphs: "Could not connect to database: unable to open database file." Additionally, some modules in the Shiny interface are not fully implemented. Considering that software updates and optimizations require continuous user feedback, the authors can address potential demands in future versions. However, for the functionalities mentioned in the manuscript, the authors should strive to fix these obvious bugs. Furthermore, it is recommended that the authors add a reset option in the Subset module, allowing users to return to the original initial dataset.Is installation/deployment sufficiently outlined in the paper and documentation, and does it proceed as outlined?YesAdditional CommentsAlthough the authors have provided installation documentation, the current documentation on GitHub is not user-friendly. For example, the page at https://github.com/cobriniklab/chevreul does not include code for importing seuratTools, yet it runs the built-in function `clustering_workflow` from seuratTools. Additionally, the current documentation is overly simplistic and not accessible to those without programming experience.Is the documentation provided clear and user friendly?NoAdditional CommentsThe authors have separated the example workflows for SingleCellExperiment objects and Seurat objects into two different GitHub projects, which is not conducive for users to understand the structure of Chevreul or to facilitate learning. Additionally, the batch integration mentioned in the article lacks specific implementation examples. The authors should at least provide implementation examples for the results mentioned in the manuscript. Furthermore, the current documentation needs further refinement to truly enable individuals without programming expertise to easily analyze single-cell data.Is there enough clear information in the documentation to install, run and test this tool, including information on where to seek help if required?YesAdditional CommentsIs there a clearly-stated list of dependencies, and is the core functionality of the software documented to a satisfactory level?NoAdditional CommentsThe authors have developed an excellent Shiny app for single-cell visualization, enabling users without programming expertise to freely export visualization results from single-cell analysis. The installation commands provided by the authors on https://github.com/cobriniklab/chevreul do indeed allow for the installation of Chevreul. However, Chevreul involves nearly 300 dependency packages, including sub-libraries developed by the authors (seuratTools, chevreulPlot, chevreuldata, chevreulPlot, chevreulProcess, chevreulShiny) as dependencies. Relying solely on the installation commands provided by the authors to install all dependency packages may result in some packages (especially large ones) failing to install due to network bandwidth issues, which is not user-friendly for those without programming experience. Additionally, could the numerous dependency packages of Chevreul potentially cause dependency conflicts with existing R environments? Should the authors recommend users to deploy Chevreul in a new R environment? It is recommended that the authors provide a step-by-step installation guide, explaining potential issues and solutions during the installation process based on the dependencies of Chevreul and its sub-libraries. By installing dependency packages step by step, users can gradually complete the installation of Chevreul. The current installation documentation is clearly not user-friendly for non-programmers and does not align with the authors' statement in the manuscript: "It differs from other scRNAseq analysis packages in its ease of installation and use." At present, the installation documentation provided by the authors may not meet the original design intent of Chevreul. Additionally, the authors should specify that Chevreul supports Seurat version V5.Have any claims of performance been sufficiently tested and compared to other commonly-used packages? NoAdditional CommentsThe authors could provide specifications for the minimum hardware requirements needed to run Chevreul, such as the number of CPU cores and the amount of memory. Additionally, the authors could offer data on the runtime of Chevreul as the volume of data increases.Is test data available, either included with the submission or openly available via cited third party sources (e.g. accession numbers, data DOIs)?Additional CommentsAre there (ideally real world) examples demonstrating use of the software? YesAdditional CommentsIs automated testing used or are there manual steps described so that the functionality of the software can be verified?NoAdditional CommentsAny Additional Overall Comments to the AuthorThe authors have developed an R Shiny app for single-cell exploratory data analysis, which will significantly expand the application scenarios of single-cell data analysis and bring great benefits to a wide range of biology practitioners. The large size of Chevreul's installation package indicates the considerable difficulty in its development, reflecting the immense wisdom and effort the authors have invested in creating this package. Chevreul's advantages in visualization and analysis are evident, and if further developed and refined, it is certain to attract even more users in the future. To ensure that such an excellent package as Chevreul can be easily and quickly adopted by users, several suggestions for improving the documentation and enhancing user-friendliness are provided. We hope the authors can refine the package based on the reviewers' feedback and recommendations.RecommendationMajor Revisions